# Desire for Genital Surgery in Trans Masculine Individuals: The Role of Internalized Transphobia, Transnormativity and Trans Positive Identity

**DOI:** 10.3390/ijerph19158916

**Published:** 2022-07-22

**Authors:** Annalisa Anzani, Marco Biella, Cristiano Scandurra, Antonio Prunas

**Affiliations:** 1Department of Psychology, University of Milano–Bicocca, Edificio U6, Piazza dell’Ateneo Nuovo 1, 20126 Milan, Italy; antonio.prunas@unimib.it; 2Psychologisches Institut, Ruprecht-Karls-Universität Heidelberg, 69117 Heidelberg, Germany; marco.biella@psychologie.uni-heidelberg.de; 3Department of Neuroscience, Reproductive Sciences and Dentistry, University of Naples Federico II, 80138 Naples, Italy; cristiano.scandurra@unina.it

**Keywords:** transnormativity, trans positive identity, internalized transphobia, gender-affirmation surgery, gender dysphoria, social dysphoria

## Abstract

Some trans people experience gender dysphoria, which refers to psychological distress that results from an incongruence between one’s gender assigned at birth and one’s gender identity. People who are trans masculine or nonbinary assigned-female-at-birth may pursue multiple domains of gender affirmation, including surgical affirmation (e.g., masculine chest reconstruction, penile reconstruction, etc.). The present study aimed to investigate the possible factors involved in trans people’s desire to undergo gender-affirming genital surgery. Trans masculine and nonbinary participants (*N* = 127; mean age = 26.90) were recruited through a web-based survey and completed self-report instruments (i.e., the Internalized Transphobia subscale of the Gender Minority Stress and Resilience Measure, the Trans Positive Identity Measure, the Gender Congruence and Life Satisfaction Scale, an ad hoc scale on transnormativity, and a single-item on desire to undergo genital affirmation surgery). A path analysis showed that higher levels of internalized transphobia led to more significant genital discomfort via a dual parallel mediation of transnormativity and positive identity. Moreover, this genital discomfort fueling pattern was the most significant predictor of the desire to undergo genital surgery as the effect of internalized transphobia was fully mediated by increased genital discomfort. Findings are discussed in the light of the recent strand of research on gender dysphoria as a multifaceted construct, with social components.

## 1. Introduction

“Trans” is a term referring to people whose gender identity, expressions, or behaviors differ from those typically associated with their gender assigned at birth [1]. This group of people includes a wide range of gender identities, usually categorized in binary (i.e., male-to-female or female-to-male trans people) or nonbinary (i.e., trans individuals who do not identify themselves into a binary view of gender, such as genderqueer, bigender, etc.). 

Some trans individuals experience different degrees of gender dysphoria (GD) due to the incongruence between their body and their perceived gender and seek medical affirmation treatments to alleviate their body dysphoria. However, psychiatric models usually emphasize the role of anatomy, not considering the role of certain psycho-social dimensions as potential variables that could push trans people to need to undergo medical affirmation interventions [2]. To this end, within the minority stress model [3], Lindley and Galupo [2] demonstrated that GD may be understood as a proximal minority stressor due to its interaction with both external triggers and mental health symptoms. This means that GD—and, with it, the need to undergo medical affirmation interventions—may be also associated with contextual factors that are internalized by trans individuals, such as internalized transphobia, transnormativity, and trans positive identity. 

Thus, within a psycho-social approach to GD, the current study aimed at assessing the role of transnormativity and trans positive identity as potential mediators in the relationship between internalized transphobia and genital discomfort. In the following paragraphs, we will first provide an overview on GD, highlighting the shift from a medicalized to a psycho-social perspective. Then, we will introduce the psycho-social variables considered in this study (i.e., internalized transphobia, transnormativity, and trans positive identity) highlighting their potential relationships with GD and the need to undergo medical treatments.

### 1.1. The Crisis of the Clinical Perspective on Gender Dysphoria

GD is a term that has taken on different meanings in its use by the trans community, in the psychological and psychiatric literature, and within political and public discourse. Generally, GD refers to the distress experienced when one’s own gender identity is different from the gender assigned at birth [2]. However, since the 5th edition of the Diagnostic and Statistical Manual of Mental Disorders (DSM-5) [4], this term which was already well used by the gender-diverse population and within the academic discourse to describe an experience of distress has become a diagnostic label [5]. 

Traditionally, GD has been described by clinical and psychiatric perspectives according to a binary, medicalized, and body-centered conception. These clinical lenses mostly emphasized body dissatisfaction and incongruence [6]. These models were particularly constructed based on clinicians’ interpretation of trans women’s experiences, by describing dysphoria as the experience of being born in the “wrong” body (i.e., the male body) and created typologies of trans women based on their sexual orientation, conflating sexual and gender identity (i.e., [7]). 

For the past few years, the DSM-5 diagnosis of GD has been strongly criticized, like the very presence of the condition in a psychiatric manual. The primary focus of this criticism is the idea that trans persons are mentally disturbed simply because they identify as nonconforming or transgender. Trans people have a long history of psychiatrization, where mental health professionals stood (and in some cases still do) as gatekeepers of the medicalized transition journeys. As a result, the definition of the trans identities as a psychiatric condition has also had long-term consequences for how we have taken charge at the socio-health level of trans people’s transitions [8,9,10]. The APA defines GD as the “distress that may accompany the incongruence between one’s experienced or expressed gender, and one’s assigned gender” ([4], p. 451). However, several limitations and contradictions in this diagnosis have been highlighted [5]. One of the most controversial points is the focus on distress, which would have made the GD diagnosis less pathologizing in the intentions of the APA task force. Nonetheless, Davy and Toze [5] noted that the DSM diagnosis also captures conditions in which individuals do not experience distress due to experiencing gender incongruence ([4], p. 451)—factually nullifying the justification for including the diagnosis in a psychiatric manual. 

The psychiatric definition of GD has also been criticized for its excessive focus on anatomy and reduced emphasis on self-identification [11,12]). Indeed, the DSM-5 diagnostic criteria (i.e., “A marked incongruence between one’s experienced/expressed gender and primary and/or secondary sex characteristics”), as well as the scales used to measure GD, are strongly focused on the body (e.g., the Utrecht Gender Dysphoria Scale [UGDS]; [13]). A recent study asked a group of people self-identifying as transgender to rate the extent to which clinical measures of GD captured their experience. In general, the scales were rated as not very representative of trans people’s experience of GD. Only little more than 50% of participants rated the UGDS and the Gender Identity/Gender Dysphoria Questionnaire for Adolescents and Adults [14] as descriptive of their experience. Moreover, the scales were rated more positively by transfeminine individuals, followed by transmasculine and nonbinary people, as might have been expected given that diagnostic classifications were made from clinical observation of trans women [6]. Qualitative research has been beneficial in allowing the experience of trans people to be put back at the center, giving them a first-person voice in narrating their subjective experience of GD. Qualitative studies also allowed to better nuance the phenomenological description of GD, drawing in “classical” conceptualizations of GD, retaining the parts that are reflected in trans people’s experience, but also expanding from traditional conceptualizations by going beyond the limits of the clinical perspective of GD [6]. Experienced incongruity with one’s own body is an essential part of the trans experience, aligning with the traditional clinical perspective. In the study by Pulice-Farrow et al. [6], participants highlighted how GD manifested as a feeling of disconnection from their body or appearance. Furthermore, this feeling of disconnection between one’s gender identity and the body was often the cause of the distress or discomfort reported by participants (consistent with the DSM-5) [6]. On the other hand, the narrative of GD includes many other nuanced experiences that fail to fit into the clinical model. For example, dysphoria may not manifest as a rejection of one’s “assigned” gender characteristics, but rather include experiences in which individuals do not possess the desired balance between femininity and masculinity. Moreover, the clinical model contemplates a fixed and constant experience of dysphoria, which may be alleviated as a result of a transition journey (social, medical, and legal). The experience reported by people tells something different. Dysphoria may be variable over time and in different social contexts [6]. Trans people do not necessarily take linear paths to transition, but the paths are individualized [15].

In light of the limitations of the clinical conceptualization of GD, a recent model proposes a conceptualization of dysphoria that recognizes its social components and emphasizes the self-identification aspects over the focus on anatomy. Lindley and Galupo’s [2] theoretical model proposed to include GD among the proximal stressors of the minority stress model. In recent years, indeed, it has been recognized that there is a significant social component to gender dysphoria that social situations can trigger, e.g., misgendering or questioning one’s identity [16,17]. The authors demonstrated how GD is an intermediary between external triggers (or distal stressors) and health outcomes (e.g., mental health symptoms). In short, the experience of GD involves the internalization of social experiences, and this would broaden the clinical perspective that primarily recognizes bodily distress [2].

In the present paper, we approach gender dysphoria as multifaceted, by considering both its social and body discomfort components.

### 1.2. Gender Dysphoria, Desire for Medical Affirmation Treatments, and Psycho-Social Constructs

Considering the previous discourse about the clinical approach to GD, it is not surprising that the medical gender affirmation journey has long been viewed as the only way to alleviate distress from GD (i.e., [18]). Transitioning was considered a linear and binary path [19]. Indeed, the effectiveness of hormonal and surgical gender affirmation treatments appears to lead to good mental health and well-being outcomes for people who seek them [20]. However, this approach, which reifies anatomic discomfort and medical transition, excludes a substantial proportion of people who identify as trans or nonbinary as not all trans individuals experience body dysphoria or seek medical affirmation treatments [21,22]. 

Conceptualizing dysphoria as a proximal stressor can help move beyond a strictly anatomical view by recognizing its social aspects. Indeed, the extent to which individuals experience gender congruence is influenced by the nature of their beliefs about trans identity. For example, internalized transphobia influences the level of experienced GD [2,23,24]. This evidence highlights how an experience considered strictly individual and related to discomfort with one’s own body (that is GD), could be influenced by the internalization of negative beliefs about trans identity that are widespread and reinforced by the social context. 

In the current paper, we investigated the possible association of societal factors on trans people’s desire to undergo genital affirmative surgery, via genital discomfort. In particular, we investigated the role of transnormativity and trans positive identity.

Transnormativity could be defined as the social rules and expectations placed on trans people [25]. It may otherwise be defined as a sort of pressure to be “normalized” as cisgender, by outlining the ways of being trans that are considered legitimate, and thus establishing a hierarchy of ways of having a trans identity that are more acceptable than others. To this end, Bradford and Syed [25] conducted 4 focus groups that involved 15 transgender U.S. residents. The authors highlighted the components of transnormativity from the narratives of trans individuals, who described expectations with respect to (1) the medicalization of their bodies; (2) a preference for binary identities; (3) a pressure to conform to a gender role; (4) a predilection for trans identities manifesting very early in life; (5) the victimization of trans identities; (6) the reification of transnormativity in legal systems and social structures through gatekeeping; and (7) the legitimacy of certain narratives and trans identities. These individual aspects fit together as mutually reinforcing pieces of an underlying coherent narrative rather than as independent stereotypes. Thus, since one aspect of transnormativity is fundamentally linked to the others, the reinforcement of each individual aspect serves to reinforce the entire narrative [25]. In this sense, it is fair to say that there is a hierarchy of trans identities at the top of which are those individuals who are as close as possible to cisgender identity, delegitimizing the experience of trans people who do not pursue medicalization, who have a gender expression that is outside of the binary (i.e., not clearly feminine or masculine), who reject pressures to conform to a specific gender role, or who discover their trans identity in adulthood or later life. 

The notion of positive trans identity, instead, describes something very different. Although often only the most negative aspects of trans experience and outcomes are highlighted, positive experiences can play an important role in terms of well-being, personal growth, resilience, and social support [26,27,28]. Riggle et al. [29], through an online survey collecting data on self-reports of the positive aspects of a trans identity (*n* = 61), identified eight themes of positive trans identity that are related to positive outcomes in terms of individual’s strengths and resources, as follows: (1) a feeling of congruence between one’s inner self and outer self or expression; (2) improved interpersonal relationships through acceptance by family and friends; (3) personal growth and resilience (confidence, strength, and self-awareness); (4) greater empathy and sensitivity to others; (5) a unique perspective on both genders through personal experiences; (6) living beyond the gender binary and challenging gender norms and stereotypes; (7) committing to activism and educating others; and (8) connecting to LGBTQ communities. We might hypothesize that positive trans identity, as opposed to internalized transphobia, levers aspects of individual and community resilience and resources that are linked to a globally positive view of one’s trans identity. This globally positive attitude might also be linked to a greater acceptance of one’s body. With respect to these latter variables, their link to experienced GD has not yet been studied in the literature.

### 1.3. The Current Study

Given the link between internalized transphobia and the experience of GD already reported in the literature [2,23,24], the current study aimed at assessing the mediating role of transnormativity and positive identity in the relationship between internalized transphobia and the desire to undergo genital surgery.

We hypothesized that, as the levels of internalized transphobia increase, the endorsement of transnormative social expectations might vary in the same direction as well, while the levels of trans positive identity might vary in the opposite direction. We expect both effects to be associated with the levels of perceived GD, particularly genital discomfort. Therefore, we expect them to have a subsequent effect on the perceived need to undergo genital surgery for gender affirmation.

## 2. Materials and Methods

### 2.1. Participants and Procedures

Participants were recruited through a web-based survey as part of a larger study aimed at investigating different aspects of the experiences of trans masculine individuals with masculinity. [30] The study’s inclusion criteria were: (1) being a trans masculine person; (2) being at least 18 years old; (3) and living in Italy for at least 10 years. Recruitment entailed identifying groups on social media (Facebook and Instagram), whose members might be interested in participating in a study, dealing with topics related to the LGBTQ+ community or specifically to the trans community. In addition, Italian stakeholders in the trans community were contacted and asked to disseminate the survey through their listservs, activating a snowball sampling recruitment procedure. Completing the study took an average of 15 to 20 min.

Trans masculine and nonbinary people (who identify in a masculine spectrum, this indicates that the research includes individuals who self-identify to some extent as masculine despite being nonbinary) were included in the study. The study was conducted entirely in Italy and in the Italian language. No incentives were offered for participation.

Two-hundred and eleven participants clicked on the questionnaire link and, among them, 175 agreed to the informed consent form. Of the sample of 175 participants, 6 were excluded because they were not assigned female at birth. Among the 169 participants, 43 did not answer at least one scale of interest to the study. Thus, the final sample includes 127 individuals. The participants’ gender identities were recorded by asking them to write down the label they use to define their gender, but also with a close-ended question asking them which gender would represent them the most. Altogether, 88.2% of the sample identified as trans men, 11.8% as nonbinary. The sample included individuals at different stages of social and medical transition (see Table 1). Other sociodemographic characteristics are included in Table 2.

The Institutional Review Board approved the study at University of Milano-Bicocca with the ethical code: RM 2021-637. Informed consent was obtained from all individual participants included in the study.

### 2.2. Measures

*Socio-demographic characteristics.* Demographic information included gender identity, age, and educational level, sexual orientation, marital status, relationship status and social and medical transition steps. Participants provided their gender identity as a write-in response and then selected their primary gender identity from two discrete options (trans men and nonbinary). As for sexual identity, participants provided their sexual identity as a write-in response and then selected their primary sexual orientation among different discrete options (asexual, bisexual, fluid, gay, heterosexual, pansexual, queer, other). 

*Internalized transphobia*. Internalization of negative beliefs about trans identity was measured through the Internalized Transphobia (IT) subscale of the Gender Minority Stress and Resilience Measure [31,32]. Responses range from 1 (=strongly disagree) to 5 (=strongly agree). A sample item is “I often ask myself: Why can’t my gender identity or expression just be normal?”. The scale demonstrated good internal consistency (Cronbach’s α = 0.89).

*Trans positive identity*. The Trans Positive Identity Measure (T-PIM; [28]) is a 24-item self-report scale that measures positive experiences with one’s trans identity in relation to five domains (authenticity, intimacy/relationships, belonging to the trans community, social justice/compassion and insight/self-awareness). Responses ranged from 1 (=strongly disagree) to 7 (=strongly agree). A sample item is “I embrace my trans identity”. The scale showed good internal consistency (α = 0.89). The Trans positive Identity Measure used in this study was not previously validated in Italy. For this reason, it has been translated into Italian following all the phases suggested by Behling and Law [33] related to back-translation procedures.

*Transnormativity.* Transnormativity was measured through items developed using qualitative data from the study by Bradford and Syed [25], which investigated transnormativity within a sample of trans and non-binary individuals. The authors identified 7 themes (as discussed in the introduction) that reflect how society defined the “normative” trans person. Based on their findings, we developed an 18-item scale to assess the extent to which participants adhere to these normative beliefs and expectations about trans identities. Responses ranged from 1 (=strongly disagree) to 5 (=strongly agree). The scale demonstrated a good internal reliability, Cronbach’s α = 0.91. All the scale’s items are reported in Appendix A. 

*Genital Discomfort.* The Gender Congruence and Life Satisfaction scale [34] asks respondents to think about how they have felt over the last 6 months and to rate their responses on a 5- point Likert scale. Responses ranged from 1 (=never) to 5 (=always). The subscale on genital discomfort includes 6-items that pertain to distress and incongruence relating to the genitals. An example item is “I have felt that genital surgery will address the unhappiness I experience in relation to my gender”. The scale demonstrated a good internal reliability (α = 0.90). This scale has also been adapted with the use of back translation. 

*Desire for Genital Surgery*. Desire for genital surgery was assessed by a single-item question, that was: “Have you undergone (or plan to undergo) genital surgery to affirm your gender identity?” Participants had as response options: (1) “It does not apply to me/I am not interested”, (2) “I have already done it”, and (3) “I wish to do it in the future.” Participants who had already undergone genital surgery were excluded from analysis. Thus, responses were categorized as dichotomous. 

### 2.3. Data Analysis

All statistical analyses were performed using SPSS version 26 or R software, setting the level of significance at 0.05. Correlation between variables are summarized in Table 3. The main hypotheses of the current study were tested by running a path analysis using the R package lavaan (estimated using diagonally weighted least squares given the binary nature of the Desire for Genital Surgery). The model tested contained Internalized Transphobia as an exogenous variable which directly affected all other variables. It was expected that Internalized Transphobia would predict Genital Discomfort, and that its effect would reach such a variable via a double parallel mediation through Transnormativity and Trans Positive Identity. Specifically, in the model our exogenous variable predicted the two mediators which were entered as predictors of the discomfort for genitals. Additionally, we tested whether these variables could predict desiring genital surgery. To do so, we extended our model allowing Genital Discomfort and Internalized Transphobia to directly predict whether the participant desires going through genital surgery. Given that Desire for Genital Surgery was a binary variable, we accommodated it via a probit regression allowing us to estimate the probability that our participants were willing to go through the surgical procedure of interest. Moreover, given the design of the model, we were able to test whether the effect of Internalized Transphobia on the Desire for Genital Surgery was mediated by Genital Discomfort. Additionally, we tested whether the state of advancement of the medical transition affected our main path analysis by running a multi-group confirmatory factor analysis, splitting the sample in two groups: those who did not perform any medical transition step versus those who performed at least one step. If the main path analysis held across both groups, the effect of the grouping variable (i.e., the level of medical transition) could be excluded. 

## 3. Results

Regarding the parallel mediation of Internalized Transphobia on Genital Discomfort via Transnormativity and Trans Positive Identity, our analysis confirmed the expected pattern (see Figure 1). Specifically, results suggested that the effect of the exogenous variable on the target variable was partially mediated by both mediators, as the coefficient of Internalized Transphobia was significant despite the presence of the mediators, *b* = 0.25, *z* = 2.35, *p* = 0.019. 

Moreover, the indirect effects were significant for both Trans Positive Identity, *b* = 0.10, *z* = 2.89, *p* = 0.004, and Transnormativity, *b* = 0.08, *z* = 2.23, *p* = 0.026. These results suggested that increased levels of Internalized Transphobia were directly linked to increased levels of Genital Discomfort. 

Additionally, an increase in Internalized Transphobia was likely to induce both an increase in Transnormativity, which in turns induced greater Genital Discomfort, and a decrease in Trans Positive Identity, which in turns tended to increase Genital Discomfort. In other words, our results showed that increased levels of Internalized Transphobia were related to increased levels of Genital Discomfort both directly and indirectly via the two mediators (Transnormativity and Trans Positive Identity). Moreover, our analyses suggested that Internalized Transphobia affected the desire for surgery via increased Genital Discomfort. The pattern we found is indicative of a total mediation as the direct effect of the exogenous variable on Desire for Genital Surgery was not significant if Genital Discomfort was introduced in the model, *b* = −0.18, *z* = −1.18, *p* = 0.238. The latter, however, was the only significant predictor of the Desire for Genital Surgery, *b* = 0.86, *z* = 5.13, *p* < 0.001, suggesting that greater genital discomfort was linked to increased desire for surgery. 

The total mediation pattern was confirmed by a robust indirect effect of Internalized Transphobia on Desire for Surgery via Genital Discomfort, *b* = 0.37, *z* = 3.446, *p* = 0.001.

To prove that our findings were robust to the level of medical transition experienced by our participants, we fitted a multi-group confirmatory factor analysis by constraining several parameters to be equal across groups and comparing if the model with such constraints remained satisfactory [35]. Results suggested that, even with the most restrictive constraints across groups (intercepts and loadings), the fit of the model was not disrupted, *χ*^2^ = 5.94, *p* = 0.139. This confirmed that the level of medical transition experienced by our participants did not affect the pattern of results derived from the main path analysis. 

Taken together, these results suggested that people with greater internalized transphobia exhibited greater genital discomfort due to increased transnormativity and decreased trans positive identity. Moreover, this genital discomfort-fueling pattern was the only responsible for the increased desire for surgery, as internalized transphobia could not explain the target construct when genital discomfort was in the model. Additionally, all results held even when controlling for the stages of the participants’ medical transition. The main path analysis held across both groups considered (those who did not perform any medical transition step vs. those who performed at least one step). Therefore, we can conclude that effect of the grouping variable (the level of medical transition) could be excluded.

## 4. Discussion

Starting from the recent evidence highlighting the social components of GD, in the present work we investigated which factors are associated with the level of perceived incongruence to one’s genitals and, ultimately, the desire to undergo gender affirmative surgery. In particular, we focused on the potential impact of psycho-social variables. Our results should be considered within the Italian transgender population environment, whose contexts they live in cannot be considered as highly supportive (e.g., [36]). For example, the Italian law 164, which regulates Gender Affirming Surgery (GAS), was promulgated in 1982 and has never been updated. Currently, it is a court to determine whether a trans person can undergo GAS [36]. Furthermore, Italy is not only lagging in terms of policies related to gender transition, but also for anti-discrimination or LGBTQ+ civil rights policies. Our findings demonstrated that social components may play a role in the perceived discomfort with the body (and genitals in particular) in trans masculine individuals, and ultimately in the desire to undergo gender affirmative surgery. It is important to emphasize that our results did not highlight a direct relationship between psychosocial variables and actual genital surgery. In other words, the participants’ current desire for genital surgery was not necessarily related to whether they will have it in the future. The model we tested can be embedded within the gender minority stress theory [3,37]. Recent extensions of the minority stress theorization, particularly the psychological mediation framework [38] claimed that certain group-specific variables (i.e., proximal stressors, such as internalized transphobia), and general psychological processes mediate the relationship between distal stigma factors (such as violence and discrimination) and mental health. Our study specifically explored the relationship between internalized transphobia, one of the most significant proximal stressors, and gender dysphoria (particularly genital dysphoria). The relationship between internalized transphobia and dysphoria seemed to be partially explained by two intermediate constructs: transnormativity (that increased the level of discomfort for one’s genitals) and trans positive identity (that promoted a positive relationship with one’s genitals). 

Transnormativity was shown to be linked to high internalized transphobia and to partially explain an increase in the level of genital discomfort. Transnormativity is a normative ideology that holds trans people’s experiences and identities accountable to a binary, medical framework [39]. According to this belief system, the legitimacy of trans people’s identities is socially evaluated, and trans individuals are rewarded or sanctioned depending on how closely their experience aligns with these normative standards (i.e., medicalized person, who easily “passes” for a cisgender person and looks as conforming as possible to gender stereotypes). As Bradford and Johnson [39] have pointed out, transnormativity has probably a relevant impact on the lived experience of trans people in several domains of life, although research has only marginally explored it. Our study is the first to explore the connection between transnormativity and genital discomfort, which can increase due to bodily characteristics that distance the person from that ideal, transnormative model. It is important to keep in mind that, as transnormativity permeates all health disciplines, trans people’s bodily autonomy within health care interactions can be very limited [39]. On the other hand, valuing the unique characteristics that define one’s trans identity, i.e., having high levels of trans positive identity, showed the opposite effect, increasing genital acceptance and ultimately influencing the desire for having gender-affirming surgery. Therefore, the present study could be integrated with the most recent literature demonstrating that GD is a complex, multifaceted construct with important social components that should not be overlooked. However, the present study is not without limitations. First, our participants represent an online convenience sample of Italian trans masculine individuals. This prevents any kind of generalization to other trans populations, such as trans feminine individuals, or to different social contexts. Indeed, the role of cultural context is central when a study is focusing on variables with a strong social significance. However, online sampling allowed us to reach trans masculine participants from the general population, to ensure that we did not oversample participants who have undergone or are seeking medical transitions. Second, the study was cross-sectional, making the direction of the relationships between the variables examined only theoretically hypothesized. Future longitudinal research designs are needed to discern the cause-effect relationships between variables analyzed in the current study. Third, although our results suggested that social variables might influence the decision to undergo genital surgery in trans masculine persons, this does not imply that no other individual and contextual factors may contribute to decision-making. Furthermore, even if societal variables partially influence a person’s decision to undergo gender affirmation surgery, this does not mean that the decision is any less valid. Thus, it is crucial to point out that the “medical model,” centered on the individual’s experience of body discomfort, is a journey that is valid for many people [6]. Furthermore, it is worth noting that we refrained from deepening our investigation comparing potentially interesting participants’ sub-groups (i.e., based on the surgical procedure of interest). This decision is mostly based on statistical power concerns. Future research might look at the possible impact of hormone therapy for gender affirmation on this pattern. Future studies might further explore the role of gender affirming hormone therapy in larger and more balanced samples, even though our model confirmed the hypothesized pattern both on the entire sample of participants and when running a multi-group confirmatory factor analysis, splitting the sample into two groups: those who did not perform any medical transition step and those who performed at least one step.

## 5. Conclusions

By investigating the social components that might trigger GD, the present study provided researchers and clinicians with significant insights for working with trans masculine individuals. Indeed, the results presented are further evidence that corroborates recent hypotheses on the role of specific social dynamics in GD, considered until recently as an experience purely internal within the individual. These results highlighted that a perspective focused on body distress and medicalization is insufficient to understand the trans experience fully. Thus, future research will benefit from focusing on social triggers of GD, the belief system that sustains them, and how to clinically unhinge these rigid social norms. In this regard, clinical work focusing on the client’s internalized beliefs and assessing the extent to which there may be elements of internalized transphobia or transnormativity can be beneficial. Finally, trans positive identity emerged as a factor contributing to higher satisfaction and congruence with genitals. This result shed light on an area where clinicians could work beneficially with clients to develop a better relationship with one’s trans identity. Furthermore, clinicians can assist clients and their partners with developing a support network in which they might find positive and supportive role models.

## Figures and Tables

**Figure 1 ijerph-19-08916-f001:**
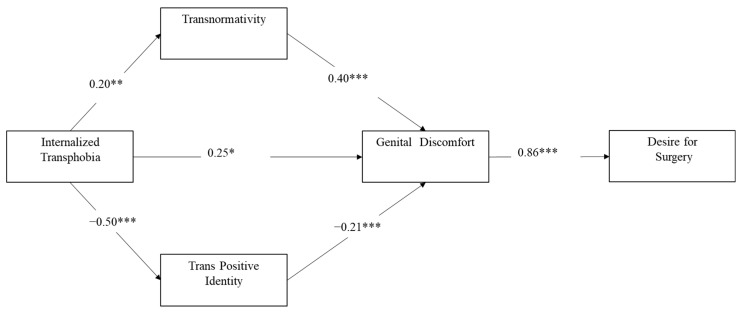
Results from the path analysis. Notes. * *p* < 0.05; ** *p* < 0.01; *** *p* < 0.001.

**Table 1 ijerph-19-08916-t001:** Transition status. Social and medicalized transition steps taken by the participants.

	Total Sample (*N* = 127) *N* (%)
**Social transition**	
Coming out to family members	102 (80.3)
Coming out to friends	119 (92.9)
Coming out with school mates/colleagues	76 (59.8)
Chose different name	115 (90.5)
Changed name legally	18 (14.2)
Wearing clothes that reflect GI in public	121 (95.3)
Wearing clothes that reflect GI at work/school	118 (92.9)
Changed gender legally	18 (14.2)
**Medicalized transition**	
Top surgery	18 (14.2)
Bottom surgery (penile reconstruction)	1 (0.8)
Voice Therapy	8 (6.3)
HRT (Testosterone)	55 (43.3)
Hysterectomy	12 (9.4)

**Table 2 ijerph-19-08916-t002:** Sociodemographic characteristic of the total sample and the sample split by binary and nonbinary identities.

	Total Sample *N* = 127	Binary *N* = 112	Non-Binary *N* = 15
**Age**	*M* = 26.90 (*SD* = 9.93)	*M* = 26.53 (*SD* = 9.91)	*M* = 29.78 (*SD* = 9.93)
**Sexual Orientation**; *n* (%)			
Asexual	3 (2.4)	1 (0.9)	2 (13.3)
Bisexual	21 (16.5)	19 (17.0)	2 (13.3)
Fluid	4 (3.1)	4 (3.6)	-
Gay	11 (8.7)	10 (8.9)	1 (6.7)
Heterosexual	36 (28.3)	36 (32.1)	-
Pansexual	28 (22.0)	24 (21.4)	4 (26.7)
Queer	10 (7.9)	7 (6.3)	3 (20.0)
Other	10 (7.9)	8 (7.1)	2 (13.3)
**Education Level**; *n* (%)			
Secondary School	18 (14.2)	17 (15.2)	1 (6.7)
High School	72 (56.7)	66 (58.9)	6 (40.0)
Graduate or post-graduate	33 (26.0)	26 (23.2)	7 (46.7)
**Marital Status**; *n* (%)			
Single	95 (74.8)	84 (75.0)	11 (73.3)
Divorced/Separated	2 (1.6)	1 (0.9)	1 (-6.7)
Cohabitant/Common-law couple	25 (15.7)	23 (20.5)	2 (13.3)
Widow	1 (0.8)	1 (0.9)	-
**Relational Status**; *n* (%)			
Committed	38 (29.9)	38 (33.9)	3 (20.0)
Married	-	-	-
Dating	8 (6.3)	8 (7.1)	-
Polyamorous Relationship	6 (4.7)	4 (3.6)	2 (13.3)
Non-consensual non-monogamy	1 (0.8)	-	1 (6.7)
Open Relationship	6 (4.7)	4 (3.6)	2 (13.3)
Single	57 (44.9)	53 (47.3)	4 (26.7)
Not interested in having a relationship	9 (7.1)	7 (6.3)	2 (13.3)

**Table 3 ijerph-19-08916-t003:** Correlations between transnormativity, internalized transphobia, trans-positive identity, and surgery procedures desire.

Scales	1	2	3	4	*M* (*SD*)	Range
1. Transnormativity	-				3.25 (0.76)	1–5
2. Internalized transphobia	0.28 **	-			2.65 (1.06)	1–5
3. Trans positive Identity	−0.43 ***	−0.33 ***	-		9.50 (1.63)	1–7
4. Surgery Procedure Desire	−0.25 **	0.14	−0.29 ***	-	0.49 (0.50)	0–1

Notes: *M* = Mean; *SD* = Standard deviation. ** *p* < 0.01; *** *p* < 0.001.

## Data Availability

Anonymized data will be made available upon reasonable request to the corresponding author.

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
