# Peer review of "Desire for Genital Surgery in Trans Masculine Individuals: The Role of Internalized Transphobia, Transnormativity and Trans Positive Identity"

_ijerph, 2022, doi:10.3390/ijerph19158916_

Round 1

Reviewer 1 Report

The manuscript with the Title “Desire for Genital Surgery in Trans Masculine Individuals: The role of Internalized Transphobia, Transnormativity and Trans Positive Identity” reports results of an online survey among N=127 trans masculine and nonbinary participants in Italy.

The topic is relevant for the target journal and special issues. The manuscript is readable.

However, there is room for improvement in several respects:

The abstract should report more details about the sample (sample type, age range) etc.

For its theoretical model, the manuscript adopts the minority stress approach. A visualization of the conceptual model would be helpful for readers.

As the study is purely correlational, no causal claims are possible. Authors correctly state this in the limitations. Hence, they should avoid causal language throughout the manuscript such as “In the current paper, we investigated the possible influence of societal factors on trans 147 people’s desire to undergo genital affirmative surgery”. Better talk about associations instead of influence. This is another example of unsupported caual language “Additionally, an increase in Internalized Transphobia was likely to induce both an  increase in Transnormativity”. Make clear throughout that you can only make claims about associations.

In the state of research, the presented study results should be contextualized some more. For example “To this end, Bradford and Syed (2019) highlighted the components of transnormativity from the narratives of trans individuals, who described”. In situations like this pls report sample type and sample size and other relevant aspects of the study that help readers to assess the quality and generalizability of the respective study.

“Participants were recruited through a web-based survey as part of a larger study” please explain recruitment in more detail for readers.

To improve readability, put some text between table 1 and table 2.

In Tables report decimals consistently, e.g. always one decimal (26.0 instead of 26)

Table 2 “Non-consensual non-monogamy” ? Do you mean “consensual non-monogamy”

Chapter 2.2. – measures should be in intalics to improve readability and visual structure.

In Data Analysis please address issues of statistical power.

Figure 1 is missing

“Our results should 329 be considered valid with respect to the Italian transgender population” This is a gross over-generalization. Based on an online convenience/self-selection sample no generalization on the population are possible.

Data availability: Why are data and materials not openly shared?

Appendix A: Translations to English are inconsistent and not precise and should be improved by a native speaker. For example, Item 12 English men in plural, Italian L’uomo in singular etc.

Author Response

  1. The abstract should report more details about the sample (sample type, age range) etc.

Thank you for your comment. We provided more details about the sample in the abstract.

  1. For its theoretical model, the manuscript adopts the minority stress approach. A visualization of the conceptual model would be helpful for readers.

Thank you for your comment. However, as the graphic representation of the minority stress model is subjected to copyrights and the model is well-known by scholars of the field, we decided to include a figure of the model we tested instead. We hope that the reviewer will agree with us.

  1. As the study is purely correlational, no causal claims are possible. Authors correctly state this in the limitations. Hence, they should avoid causal language throughout the manuscript such as “In the current paper, we investigated the possible influence of societal factors on trans 147 people’s desire to undergo genital affirmative surgery”. Better talk about associations instead of influence. This is another example of unsupported casual language “Additionally, an increase in Internalized Transphobia was likely to induce both an increase in Transnormativity”. Make clear throughout that you can only make claims about associations.

Thank you for your comment. We rephrased few sentences (see § 1.2; 1.3; 4)

  1. In the state of research, the presented study results should be contextualized some more. For example “To this end, Bradford and Syed (2019) highlighted the components of transnormativity from the narratives of trans individuals, who described”. In situations like this pls report sample type and sample size and other relevant aspects of the study that help readers to assess the quality and generalizability of the respective study.

We have added sample size and sample type for the studies by Bradford and Syed (2019) and Riggle et al. (2011).

  1. “Participants were recruited through a web-based survey as part of a larger study” please explain recruitment in more detail for readers.

More detailed information on the recruitment methods is provided. See § 2.1

  1. To improve readability, put some text between table 1 and table 2.

Thank you, the 2nd table was moved.

  1. In Tables report decimals consistently, e.g. always one decimal (26.0 instead of 26)

Thank you for you comment. Corrected

  1. Table 2 “Non-consensual non-monogamy” ? Do you mean “consensual non-monogamy”

We measured in the survey Polyamory/Consensual non-monogamy and separately NON-consensual non monogamy

  1. Chapter 2.2. – measures should be in italics to improve readability and visual structure.

Corrected.

  1. In Data Analysis please address issues of statistical power.

We acknowledge the statistical power of the current data collection, and we would like to point out that our claims are based on the models relying on the largest sample size available. It is worth noting that we refrained from deepening our investigation comparing potentially interesting participants’ sub-groups (i.e., based on the surgical procedure of interest). This decision is mostly based on statistical power concerns. As dividing our dataset in several theoretically meaningful and interesting sub-groups would have severely undermined the statistical power of our analysis, we rather remained at the aggregated level considering the whole set of participants. Albeit interesting, further analysis would have been based on unreliable (as underpowered) design and we preferred granting ourselves reliability of the statistical procedure over potential breadth for our claims. However, we are thankful for raising such a concern as it allowed us to articulate in greater detail the trade-off we had to face during exploration of the collected data. We included your observation as a limitation (see § 5)

  1. Figure 1 is missing

Thank you, we included fig 1.

  1. “Our results should 329 be considered valid with respect to the Italian transgender population” This is a gross over-generalization. Based on an online convenience/self-selection sample no generalization on the population are possible.

Yes, we agree with the reviewer, it is a poorly phrased sentence. We meant that results should be read with caution, as they are not generalizable to a broader international scenario, but they are more specific to the Italian context, which is not supportive of trans rights. We did not intend them to apply to the entire Italian population. We rephrased for clarity.

  1. Data availability: Why are data and materials not openly shared?

We provided an appendix for the “new” material we used, and we can easily share the data with those interested.

  1. Appendix A: Translations to English are inconsistent and not precise and should be improved by a native speaker. For example, Item 12 English men in plural, Italian L’uomo in singular etc.

We have corrected the mistakes.

Reviewer 2 Report

Congratulations to the authors for the interesting paper, providing further understanding regarding the social features influencing trans experience and trans decisions (beyond medical framework).

The manuscript is clear and well written.

The Introduction clearly contextualizes the topic of the manuscript and relative literature. The authors describe their research hypothesis. Maybe they could also differentiate them by H1, H2, etc., indicating expected results.

Method section is appropriate and clearly described. The authors used some validated questionnaires, that is appreciable. To measure one of the three parameters conisdered, they created a measure, and they provided the paper they draw the items from. The authors could provide a figure to represent the hypothesized model.

Results are clear, discussion is clear and interesting as well. The authors also indicate limitations of the study and future perspective.

Author Response

Thank you for your encouraging comments. We included the figure we missed. 

Reviewer 3 Report

In this paper the authors draw on research with 127 trans masculine and nonbinary people to understand which factors lead to a desire for genital surgery.  They conclude that internalized transphobia is the largest predictor, as this prompts increased genital discomfort.  Through these findings, the authors advocate for a clinical conceptualization of gender dysphoria that incorporates psycho-social components, rather than relying on the medicalized, body-centered conceptualization of gender dysphoria, which has been widely critiqued. 

I commend the authors for their rigorous data collection and their thoughtful consideration of the limits of the previous clinical frameworks understanding trans experience.  I think they are right to encourage a move away from individualizing approaches to understanding transness and towards a more structural view of the ways an oppressive, transphobic social context influences trans people’s self-conceptions and lived experiences.   

However, I am unconvinced by the authors’ argument and framing in its current form.  I offer three pieces of feedback below, in the spirit of supporting this article’s development.

  1. We know that trans people articulate transnormative narratives in clinical settings because those have been essential to accessing medical care.  Essentially, trans people learn that these are the scripts necessary to legitimize their identities to gatekeepers, and therefore learn to mobilize them pragmatically, regardless of how deeply held these narratives beliefs actually are.  It is therefore unsurprising that people who want genital surgery would consistently tell those they perceive as medical experts (doctors, researchers, etc.) that they don’t embrace their trans identity, and trans people who don’t want surgery wouldn’t feel the need to assert that transnormative narrative as strongly.  Doesn’t this the data merely reflect trans people know which narratives are expected and necessary to access trans medical care? 
  2. The authors’ primary theoretical referent is a single recent article by Bradford and Syed (2019) on transnormativity and trans identity development, but there is inadequate engagement with the extensive trans health literature.  Perhaps because this literature is not fully engaged, the authors seem to mischaracterize the critique that scholars and trans health activists have made of gender dysphoria and the pathologization of trans identity.  The core issues in the critique have to do with the framing of transness as psychologically disturbed, and the structural subordination of trans people in defining their health and shaping their conditions for accessing care.  More grounding in the literature would make for a stronger article and contribution.  I have provided 3 citations below and the bibliographies for each offer many directions for further engagement.
  3. Given that the authors position themselves as offering a more social and less individualizing framework for understanding trans health, I find it surprising that one of their conclusions is to escalate the gatekeeping that trans people face in pursuing gender-affirming care.  They  argue, “clinical work focusing on the client’s internalized beliefs and assessing the extent to which there may be elements of internalized transphobia or trans- normativity can be beneficial to ensure that the decision to undergo medical treatment is in the best interest of the client and allows them to express their authentic identity.”  In the nuanced discussion the authors point out the limited bodily autonomy trans people experience in health care settings and affirm the validity of trans people pursuing genital surgery, so this conclusion seems out of step with the rest of the paper.  It seems to me that the authors’ psycho-social conceptualization of gender dysphoria leads more towards calls for addressing societal transphobia. 

I recognize that these are significant issues to consider, but strongly encourage the authors to do so.  I see the potential for a meaningful contribution in this paper if it can be more fully developed.  

Meyer-Bahlburg, Heino F. L. 2010. “From Mental Disorder to Iatrogenic Hypogonadism: Dilemmas in Conceptualizing Gender Identity Variants as Psychiatric Conditions.” Archives of Sexual Behavior 39(2):461–76. doi: 10.1007/s10508-009-9532-4.

Hanssmann, Christoph. 2016. “Passing Torches?: Feminist Inquiries and Trans-Health Politics and Practices.” TSQ: Transgender Studies Quarterly 3(1–2):120–36. doi: 10.1215/23289252-3334283.

Sevelius, Jae M. 2013. “Gender Affirmation: A Framework for Conceptualizing Risk Behavior Among Transgender Women of Color.” Sex Roles 68(11–12):675–89. doi: 10.1007/s11199-012-0216-5.

Author Response

  1. We know that trans people articulate transnormative narratives in clinical settings because those have been essential to accessing medical care.Essentially, trans people learn that these are the scripts necessary to legitimize their identities to gatekeepers, and therefore learn to mobilize them pragmatically, regardless of how deeply held these narratives beliefs actually are.  It is therefore unsurprising that people who want genital surgery would consistently tell those they perceive as medical experts (doctors, researchers, etc.) that they don’t embrace their trans identity, and trans people who don’t want surgery wouldn’t feel the need to assert that transnormative narrative as strongly.  Doesn’t this the data merely reflect trans people know which narratives are expected and necessary to access trans medical care? 

We thank the reviewer for giving us the opportunity to discuss an interesting point. We may consider the response bias that the reviewer had anticipated if the sample had been taken in a gender clinic and solely included clinical population. The prefabricated and more binary narrative is supposed to belong to those seeking approval for surgery if the "traditional" gatekeeping approach is to be followed. But thanks to online data collection, we could attain a nonclinical sample, avoiding the possibility of misrepresenting the truth to get a certificate. I would add a couple of other points to our argument. As far as Italy is concerned to the present date, in fact, it is not mental health professionals who have the gatekeeper role but judges. In Italy the task of the mental health professional is to establish the presence of gender dysphoria, in fact the authorization for gender affirmation surgical interventions is done through a legal ruling. Additionally, the community feels less the function of the mental health professional as gatekeeper when it comes to surgeries, particularly genital surgery, for which there are relatively few clinics in Italy that deal with them and with years-long waiting lists. In Italy, paradoxically, it is the system that acts as gatekeeper by making access to genital surgery very difficult, not mental health professionals.

  1. The authors’ primary theoretical referent is a single recent article by Bradford and Syed (2019) on transnormativity and trans identity development, but there is inadequate engagement with the extensive trans health literature. Perhaps because this literature is not fully engaged, the authors seem to mischaracterize the critique that scholars and trans health activists have made of gender dysphoria and the pathologization of trans identity.  The core issues in the critique have to do with the framing of transness as psychologically disturbed, and the structural subordination of trans people in defining their health and shaping their conditions for accessing care.  More grounding in the literature would make for a stronger article and contribution.  I have provided 3 citations below and the bibliographies for each offer many directions for further engagement.

Thank you for this useful suggestion. We believe that a paragraph that clarify the core issues related to pathologization of trans identities would frame better the introduction (see § 1.1).

For the past few years, the DSM-5 diagnosis of GD has been strongly criticized, like the very presence of the condition in a psychiatric manual. The primary focus of this criticism is the idea that trans persons are mentally disturbed simply because they identify as nonconforming or transgender. Trans people have a long history of psychiatrization, where mental health professionals stood (and in some cases still do) as gatekeepers of the medicalized transition journeys. As a result, the definition of the trans identities as a psychiatric condition has also had long-term consequences for how we have taken charge at the socio-health level of trans people's transitions [8–10].”

  1. Given that the authors position themselves as offering a more social and less individualizing framework for understanding trans health, I find it surprising that one of their conclusions is to escalate the gatekeeping that trans people face in pursuing gender-affirming care.They argue, “clinical work focusing on the client’s internalized beliefs andassessing the extent to which there may be elements of internalized transphobia or trans-normativity can be beneficial to ensure that the decision to undergo medical treatment is in the best interest of the client and allows them to express their authentic identity.”  In the nuanced discussion the authors point out the limited bodily autonomy trans people experience in health care settings and affirm the validity of trans people pursuing genital surgery, so this conclusion seems out of step with the rest of the paper.  It seems to me that the authors’ psycho-social conceptualization of gender dysphoria leads more towards calls for addressing societal transphobia. 

Thank you for the useful suggestion. We agree with the reviewer that it is a poorly phrased sentence that doesn’t fit with the scope of the paper. We deleted it.

Reviewer 4 Report

I have reviewed the manuscript entitled “Desire for Genital Surgery in Trans Masculine Individuals: The role of Internalized Transphobia, Transnormativity and Trans Positive Identity.” Unfortunately, although the article is about an important subject, it contains serious deficiencies that I think cannot be eliminated. Below, there are some suggestions that I think will improve the quality of the study.

- First of all, I should point out that it is interesting that a surgeon was not among the authors in a study on genital surgery.

Abstract:

-      “The present work…” should be changed to “The present study…”

-      Please indicate the full names of the measurement tools applied in the Abstract.

Introduction:

-      The Introduction contains too much unnecessary information beyond the scope of this study. Please rewrite this section more concisely for the purpose of the study. I must say that it is quite difficult to read this way and to understand the background of the paper.

Materials and Methods:

-      “Participants were recruited through a web-based survey as part of a larger study…” More detailed information should be given about the large study mentioned here, including eliminating the risk of salamization.

-      I recommend using only one of the terms "trans men" or "trans masculine" in the entire text.

-      What does "masculine spectrum" mean? Please explain.

-      Information about the people participating in the study should be detailed. How many people were sent surveys in total? Response rate? How many of these people agreed to participate? Was there any missing information from the participants? Have exclusion criteria been established? If so, what are these criteria? If not determined, why not determined? I believe that creating a flowchart describing the process until the final number of participants would be good for the readers to understand how the participants were involved.

-      How many parts does the survey consist of? How many minutes does it take to fill? It turns out that the first part contains the sociodemographic information. How was information about sexual orientation asked in this section?

-      I think it would be appropriate to give the sociodemographic information of binary and non-binary individuals separately.

-      Information about measurement tools should be detailed. Who developed the relevant scales? Are there cut-off points? What are the lowest and highest possible scores? Are there subscales? Have there been any adaptation studies of the measurement tools to the Italian language? If so, by whom and in what year?

Results:

-      It is seen that only one of the participants had bottom surgery, 12 had hysterectomy, and 12 had top surgery (possibly mastectomy). The number of participants who underwent such a small number of surgical operations seriously affects the power of the analyzes negatively.

-      At the same time, there are a significant number of participants who receive gender affirming hormone therapy (GAHT). The effects of GAHT appear to be overlooked. It can be said that this is a serious problem.

-      Comparative analyzes of binary and nonbinary groups may also be important. Unfortunately, the number of nonbinary participants seems to be quite limited.

Author Response

- First of all, I should point out that it is interesting that a surgeon was not among the authors in a study on genital surgery.

The paper is not about surgical techniques but mainly about desire to undergo a surgery which makes it less necessary to include a surgeon among the authors. We are interested in measuring a psychological construct regarding the desire to undergo genital surgery; we are not so interested in the extent to which this desire then turns into a request for actual surgery. The actual request for genital surgery calls into question other variables that we did not control for, and had no interest in measuring for the purpose of this paper.

Abstract:

  1. The present work…” should be changed to “The present study…”

Rephrased.

  1. Please indicate the full names of the measurement tools applied in the Abstract.

Corrected.

  1. Introduction: The Introduction contains too much unnecessary information beyond the scope of this study. Please rewrite this section more concisely for the purpose of the study. I must say that it is quite difficult to read this way and to understand the background of the paper

We agree with the reviewer about the length of the Introduction. However, we believe that information reported in the introduction are necessary to provide a clear theorical framework to understand gender dysphoria and to make clear why we tested that model. Indeed, clearly stating where we stand in the debate on gender dysphoria, as well as highlighting the social components of GD, is necessary to theoretically position our paper. That is because we included “social variables”, such as transnormativity and trans positive identity, to predict a “less” social outcome, such as the distress for one’s genitals and subsequently the desire to undergo surgery. Thus, we believe it is counterproductive to cut out pieces of the introduction that are functional to the discussion of the tested model and results. We find ourselves more in agreement with reviewer 2 in saying that the introduction appropriately contextualizes the study. We hope that the reviewer will agree with us and that they can appreciate our theoretical effort.

  1. Materials and Methods:“Participants were recruited through a web-based survey as part of a larger study…” More detailed information should be given about the large study mentioned here, including eliminating the risk of salamization.

Thank you for your comment. We provided further information on the larger study mentioned in a new paragraph called data transparency. Recently, the issue of research fatigue has emerged for some specific populations, including transgender people. There is therefore a general tendency to try to get more data from a single collection in order to address this relevant problem, especially in the context such as Italy, where money for research and remuneration to participants is scarce. For a review on research fatigue see Ashley, F. (2021). Accounting for research fatigue in research ethics. Bioethics, 35(3), 270–276. https://doi.org/10.1111/bioe.12829

  1. I recommend using only one of the terms "trans men" or "trans masculine" in the entire text.

Thank you, we uniformed where appropriate.

  1. What does "masculine spectrum" mean? Please explain.

This indicates that the research includes individuals who, although being nonbinary, identify to some extent as masculine. These individuals frequently identify as nonbinary masculine or nonbinary masc. This study excludes feminine, genderless, or fluid nonbinary identities. A footnote for clarity was included.

  1. Information about the people participating in the study should be detailed. How many people were sent surveys in total? Response rate? How many of these people agreed to participate? Was there any missing information from the participants? Have exclusion criteria been established? If so, what are these criteria? If not determined, why not determined? I believe that creating a flowchart describing the process until the final number of participants would be good for the readers to understand how the participants were involved.

Thank you for your comment. We provided detailed information in the § 2.1. “Two-hundred and eleven participants clicked on the questionnaire link and, among them, 175 agreed to the informed consent form. Of the sample of 175 participants, 6 were excluded because they were not assigned female at birth. Among the 169 participants, 43 did not answer at least one scale of interest to the study. Thus, the final sample includes 127 individuals.” Given that sample selection involved exclusion only for the gender assigned at birth and the completeness of the questionnaire, we preferred to describe it rather using a flowchart, which usually illustrates somewhat more complex processes. 

  1. How many parts does the survey consist of? How many minutes does it take to fill? It turns out that the first part contains the sociodemographic information. How was information about sexual orientation asked in this section?

In addition to the measures already described, the questionnaire also addressed masculinity. References to the published paper and its purposes have been included, as mentioned in the comment above (see § Data transparency). Additional information on the duration of compilation and sexual identity measure has been included in the text.

  1. I think it would be appropriate to give the sociodemographic information of binary and non-binary individuals separately.

Thank you for your suggestion. We provided a table with sociodemographics for both the total sample and the sample split by binary/nonbinary identity.

  1. Information about measurement tools should be detailed. Who developed the relevant scales? Are there cut-off points? What are the lowest and highest possible scores? Are there subscales? Have there been any adaptation studies of the measurement tools to the Italian language? If so, by whom and in what year?

Thank you for your comment. We provided a new table (see Table 3) with correlations between variables, the mean scores, and the range. We provided in-text citations of the original version and where possible the Italian validation of the scales we used, that you can find in square brackets following the scales name (i.e., Internalized Transphobia (IT) subscale of the Gender Minority Stress and Resilience Measure [27,28]). When papers from the Italian validation of the scales could not be found (just in the case of Trans positive Identity), the scale was back translated. We included this information in-text.

  1. Results:
    1. It is seen that only one of the participants had bottom surgery, 12 had hysterectomy, and 12 had top surgery (possibly mastectomy). The number of participants who underwent such a small number of surgical operations seriously affects the power of the analyzes negatively.

We acknowledge the unbalanced nature of the data collection and we tailor our claims accordingly. We think that having only a few participants who underwent surgery is not problematic as our claims are based on the analysis on the aggregated data and no specific claim is made regarding any particular sub-group of participants (i.e., those who underwent a specific surgical intervention). The scope of our manuscript remains at the level of the broader category of people that might experience any kind of desire for surgical intervention and do not narrow down to the level of the specific surgical procedure. The rationale behind the level of our claims (and the level of analysis) is mostly based on the specific statistical power achievable with the current sample. We think that our manuscript can provide substantial contribution to the current literature without discriminating among different desires that people can have with respect of specific surgical interventions.

  1. At the same time, there are a significant number of participants who receive gender affirming hormone therapy (GAHT). The effects of GAHT appear to be overlooked. It can be said that this is a serious problem.

We are thankful to the reviewer for raising this point allowing us to explain our claims in a clearer manner. Given that many participants received gender affirming hormone therapy, we investigated the robustness of our claim controlling for this factor. Specifically, in the manuscript, we provided information regarding the model of interest controlling for the level of intervention the participants are undergoing. Refitting our models including only participants that underwent at least one step of gender affirming interventions yielded the same pattern of results of the main analysis. Therefore, we are confident that the pattern of result is unchanged regardless the inclusion/exclusion of participants depending on the advancement of their gender affirming intervention. We based our claims on the final model including all available participants as, being that the one based on the largest sample size, it is the most reliable from a statistical point of view.

  1. Comparative analyzes of binary and nonbinary groups may also be important. Unfortunately, the number of nonbinary participants seems to be quite limited.

We agree with the reviewer that comparative analysis would be interesting, but we don’t have enough nonbinary participants to perform comparisons. We discussed it as a limitation (see Conclusions).

Round 2

Reviewer 1 Report

The action letter and revision have sufficiently addressed my concerns. 

Author Response

We thank the reviewer for the time and effort spent to revise the paper

Reviewer 3 Report

While I appreciate the authors' attention to my smaller point about their recommendation to clinicians in the conclusion, I am sorry to see the authors chose not to engage with my two substantive recommendations for the improvement of this manuscript. At this point, the manuscript remains under-engaged with the relevant literature and I do not believe that the authors' data provides a sufficient basis for the claims they are making. 

Author Response

Dear Editor,

We thank you and the reviewers for the time and effort put into reviewing the paper. We tried to answer your concern as follows:

  1. As for the fourth reviewer’s concern about the possible effects of GAHT on the model tested. We tried to emphasize more in the text the analysis we conducted to verify that the model held up, precisely to check for potential interference of GATH. That is, the model confirmed the hypothesized pattern both on the total sample of participants and when running a multi-group confirmatory factor analysis, splitting the sample in two groups: those who did not perform any medical transition step versus those who performed at least one step (that meant GATH for most of the sample). The main path analysis held across both groups, thus the effect of the grouping variable (i.e., the level of medical transition) could be excluded. Nonetheless, we recognize that the results held in a nonclinical population and may be due to not having numerically balanced groups for comparisons. Thus, we included this element in the limits and future directions.

Future research might look at the possible impact of hormone therapy for gender affirmation on this pattern. Future studies might further explore the role of gender affirming hormone therapy in larger and more balanced samples, even though our model confirmed the hypothesized pattern both on the entire sample of participants and when running a multi-group confirmatory factor analysis, splitting the sample into two groups: those who did not perform any medical transition step and those who performed at least one step.

  1. Regarding the points raised by the third reviewer, these are more theoretical concerns. However, we do not agree that all three points they raised were not addressed; the only point that did not bring changes in the text was the first one. To which we extensively responded, as the reviewer's argument could hold if the sample was not a community based but a clinical sample. The other two points were the object of revisions in the introduction and conclusion, and we also gave references to find the changes we made based on their suggestions in the text. However, we can acknowledge that we added only one paragraph in the introduction based on his criticisms, but we point out that reviewer 4 already had concerns about the length of the introduction, we did not want to overstretch the scope of the paper. We hope that the editor can agree with us.

Reviewer 4 Report

I congratulate the authors for their efforts to improve the paper. I must say that in this state the paper is definitely in a much better condition.

However, I think a few points should be re-evaluated.

1. I still agree that the Introduction part is long. (It makes up more than a third of the entire article, and it appears to be a structural issue). A long and detailed Introduction can be boring for readers.

2. I must say that my criticism of the overlooked effects of GAHT was not answered convincingly.

Here is a portion of the authors' response:

"Refitting our models including only participants that underwent at least one step of gender affirming interventions yielded the same pattern of results of the main analysis. Therefore, we are confident that the pattern of result is unchanged regardless the inclusion/exclusion of participants depending on the advancement of their gender affirming intervention."

What do the authors think about whether GAHT (or any top/bottom surgery) has an effect on "internalized transphobia", "transnormativity" and "trans positive identity"? At least this situation can be mentioned in the discussion (especially in the limitations).

Author Response

(The authors gave the same response as above.)
